# Erythroid lineage chromatin accessibility maps facilitate identification and validation of NFIX as a fetal hemoglobin repressor

Mudit Chaand [1✉], Chris Fiore [1], Brian Johnston[1], Anthony D'Ippolito [1], Diane H. Moon[1], John P. Carulli[1] & Jeffrey R. Shearstone [1,2]

Human genetics has validated de-repression of fetal gamma globin (*HBG*) in adult erythroblasts as a powerful therapeutic paradigm in diseases involving defective adult beta globin (*HBB*)[1]. To identify factors involved in the switch from *HBG* to *HBB* expression, we performed Assay for Transposase Accessible Chromatin with high-throughput sequencing (ATAC-seq)[2] on sorted erythroid lineage cells derived from bone marrow (BM) or cord blood (CB), representing adult and fetal states, respectively. BM to CB cell ATAC-seq profile comparisons revealed genome-wide enrichment of NFI DNA binding motifs and increased *NFIX* promoter chromatin accessibility, suggesting that NFIX may repress *HBG*. NFIX knockdown in BM cells increased *HBG* mRNA and fetal hemoglobin (HbF) protein levels, coincident with increased chromatin accessibility and decreased DNA methylation at the *HBG* promoter. Conversely, overexpression of NFIX in CB cells reduced HbF levels. Identification and validation of NFIX as a new target for HbF activation has implications in the development of therapeutics for hemoglobinopathies.

[1] Syros Pharmaceuticals, Cambridge, MA, USA. [2] Scientific and Medical Writing Partners, Cambridge, MA, USA. ✉email: mchaand@syros.com

Red blood cells carry oxygen using hemoglobin, which is encoded by genes that are developmentally regulated during the embryonic, fetal, and adult stages of life[3]. The fetal-to-adult hemoglobin switch, in which HbF is replaced by adult hemoglobin (HbA), is marked by silencing of fetal *HBG* and activation of adult *HBB* genes. This switch is known to be independently regulated by two transcription factors, BCL11A and LRF, that directly repress *HBG*[1,4–6]. While generally less potent, other developmental factors, lineage-defining transcription factors, and epigenetic modulators also mediate fetal globin silencing[7]. Since reactivation of the fetal form of hemoglobin in adult erythroblasts is a validated mechanism for treatment of hemoglobinopathies such as sickle cell disease[1], we set out to identify additional transcription factors critical for HbF silencing.

Previous studies to understand the regulatory networks driving the fetal and adult states of hemoglobin expression have relied on the comparison of primary cells expressing HbF, derived from CB or fetal liver, to those expressing HbA, derived from BM or peripheral blood. For example, ChIP-seq or RNA-seq has been performed on unsorted fetal and adult cells, but the developmentally asynchronous nature of this approach may have confounded the identification of lineage- and stage-specific regulators of HbF[8–10]. More recently, comparative RNA-seq profiling was performed on discrete adult and neonatal cell populations collected by fluorescence-activated cell sorting (FACS)[11]. Chromatin accessibility, as revealed by DNase I-seq and ATAC-seq, has been used to identify transcription factors driving adult erythropoiesis, but these studies lacked a fetal erythroid lineage comparator[12–14]. We built upon these previous approaches by combining cell sorting of human adult and fetal erythroid lineage cells with ATAC-seq to generate temporal chromatin accessibility profiles with the goal of identifying, and then validating, transcription factors involved in beta-like globin gene expression preference.

## Results

**Nuclear Factor One (NFI) transcription factor motifs are enriched in the adult erythroid lineage**. Using the well-established three phase erythroid differentiation culture system[15], we matured BM- or CB-derived CD34+ hematopoietic stem cells into erythroblasts and subjected them to FACS. In each lineage, we collected 7 discrete, stage-matched populations by gating based on the cell surface markers CD36 and CD235a on different days of differentiation (Fig. 1a). We further confirmed the purity of the sorted cells by flow cytometry and cytospin analyses (Supplementary Fig. 1). Erythroid cell lines HUDEP-1 and HUDEP-2[16], which predominantly express *HBG* or *HBB*, respectively, were used as controls. ATAC-seq analysis of sorted populations revealed the expected pattern of increasing chromatin accessibility at the *HBB* promoter throughout erythroid differentiation in BM-derived cells. In CB-derived cells, increasing chromatin accessibility was observed at the *HBG* promoter in populations 1–5. However, an unexpected reversion back to the *HBB* promoter was observed in populations 6 and 7, suggesting that *HBG* gene expression may be transient in CB-derived erythroblasts (Supplementary Fig. 2). Hierarchical clustering and principal component analysis (PCA) of ATAC-seq data revealed clustering of the sorted populations based on their differentiation state and not their BM or CB origin, suggesting that the majority of molecular changes during erythroid differentiation are not specific to BM or CB lineages, but rather depend on the differentiation state of the cells (Supplementary Fig. 3).

To identify transcription factors driving fetal or adult cell state, and potentially beta-like globin gene expression preference, we searched for DNA binding motifs within regions of differential chromatin accessibility. Based on clustering observed in the PCA analysis, three maturational groups were defined, populations 1–2 (early-stage erythroblasts), populations 3–5 (mid-stage erythroblasts), and populations 6–7 (late-stage erythroblasts). We found NFI motifs were significantly enriched under peaks with increased chromatin accessibility in BM relative to CB in all groups [populations 1–2, motif 1438 (FDR = $5.5 \times 10^{-3}$); populations 3–5, motifs 2339 (FDR = $2.1 \times 10^{-4}$) and 2340 (FDR = $9.6 \times 10^{-7}$); populations 6–7, motifs 1438 (FDR = $2.5 \times 10^{-5}$), 2339 (FDR = $5.9 \times 10^{-5}$) and 2340 (FDR = $4.8 \times 10^{-5}$)] (Fig. 1b, Supplementary Data 1). An orthogonal analysis approach called transcription factor footprinting showed that both flanking accessibility and footprint depth[17] at NFI motifs were also greater in BM relative to CB, providing additional evidence for increased NFI factor genome occupancy in BM versus CB cells (Fig. 1c). Supporting this observation are findings by Lessard et al., who reported genome-wide enrichment of NFI motifs in regions of differential DNA methylation in adult compared to fetal erythroblasts[18].

**NFIX chromatin accessibility and expression level are reduced in cells with elevated HbF**. Next, we compared the chromatin accessibility at the promoters of the four closely related NFI family transcription factors, NFIA, NFIB, NFIC, and NFIX[19]. A statistically significant (FDR < 0.05) increase in chromatin accessibility was observed exclusively at an *NFIX* promoter in BM-derived cells relative to their CB counterparts [populations 1 and 2, fold change = 3.5, FDR = $1.1 \times 10^{-7}$; populations 3–5, fold change = 2.3, FDR = $1.2 \times 10^{-14}$; populations 6–7, fold change = 2.9, FDR = $1.7 \times 10^{-5}$] and in HUDEP-2 cells relative to HUDEP-1 cells [fold change = 6.7, FDR = $2.2 \times 10^{-23}$; this study and Cheng et al.[20]] (Fig. 2a). Notably, this region is ~10 kb downstream of a single nucleotide polymorphism (SNP) within an *NFIX* intron linked to elevated levels of HbF in a genome wide association study (GWAS)[21] and between four differential DNA methylation clusters (25 kb upstream to 1 kb downstream) identified within *NFIX* in adult versus fetal erythroblasts[18]. Consistent with the observed chromatin accessibility difference at the *NFIX* promoter, we found that *NFIX* mRNA and NFIX protein levels were 3-10-fold higher in BM relative to CB cells and in HUDEP-2 relative to HUDEP-1 cells (Fig. 2b–d). Our data, supported by previously published work, implicated NFIX as a putative HbF repressor.

**NFIX knockdown in BM cells increases *HBG* promoter accessibility, *HBG* mRNA, F-cells and HbF**. To test if NFIX represses HbF, we lentivirally delivered short hairpin RNAs (shRNAs) to knockdown NFIX in BM and HUDEP-2 cells, leading to a 90% reduction in NFIX mRNA and protein levels (Fig. 3a, Supplementary Fig. 4). NFIX knockdown cells exhibited a slight delay in erythroid maturation, but cells still developed into reticulocytes by day 14 of erythroid culture (Supplementary Fig. 5). ATAC-seq of BM NFIX knockdown cells showed decreased chromatin accessibility at the *HBB* promoter [fold change = 2.3 and FDR = $3.9 \times 10^{-3}$] and increased chromatin accessibility at the *HBG* promoter relative to the control [fold change = 2.8 and FDR = $1.6 \times 10^{-3}$] consistent with an adult-to-fetal hemoglobin switch (Fig. 3b). DNA methylation is an epigenetic modification associated with *HBG* silencing in BM cells[22]. Following NFIX knockdown, we observed a time-dependent loss of cytosine methylation at six CpGs within the *HBG* promoter, with the largest loss at CpG −162 (Fig. 3c), which has been proposed as a biomarker for *HBG1/2* promoter activity[23].

Consistent with increased *HBG* promoter accessibility and decreased methylation, NFIX knockdown in BM cells led to a

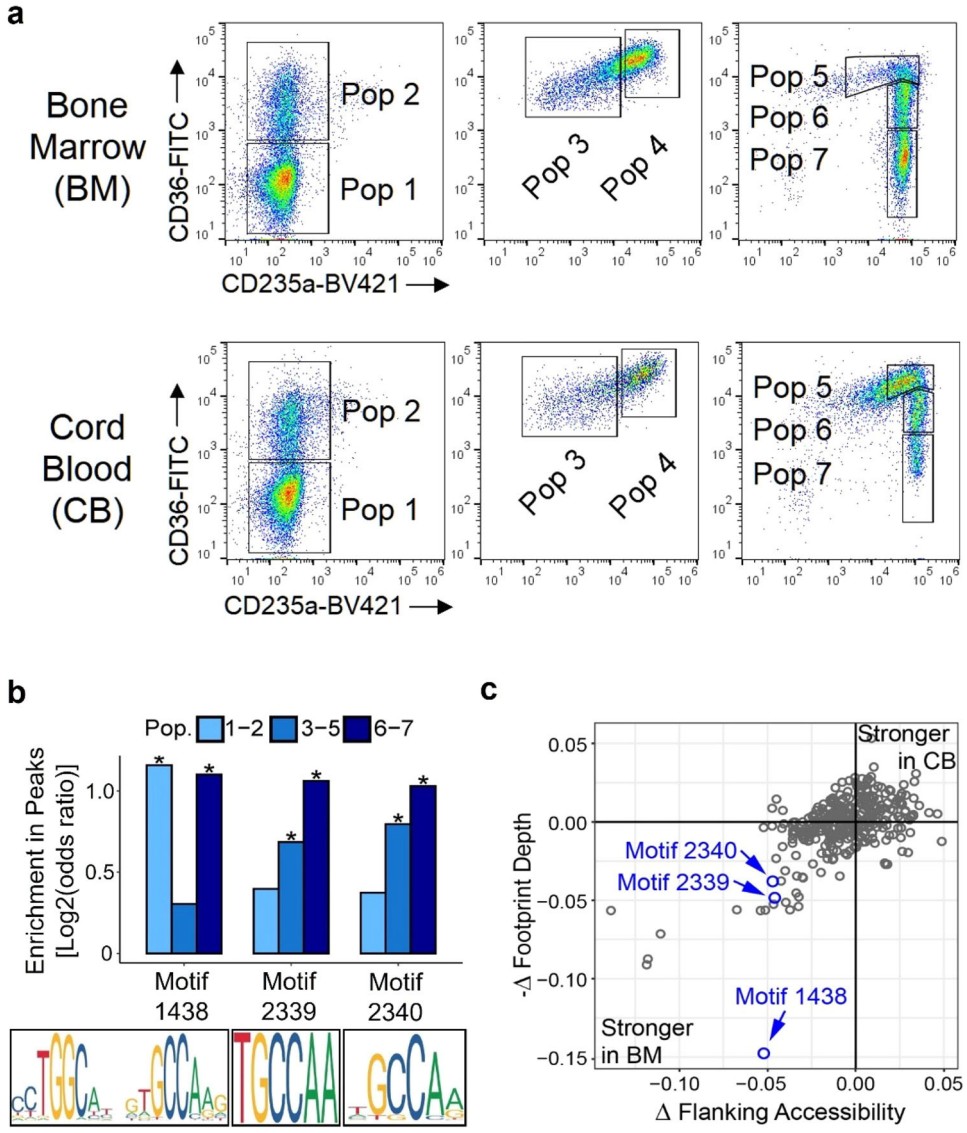

**Fig. 1 NFI factor motifs are enriched in adult erythroblasts. a** Sorting schema for primary erythroblasts derived from CD34 + BM and CB cells to obtain seven discrete cell populations based on the expression of erythroid surface markers CD36 and CD235a. **b** Analysis of ATAC-seq peaks with increased chromatin accessibility in BM-derived cells relative to CB-derived cells showing enrichment of three NFI factor DNA binding motifs numerically annotated as 1438 [TGGCANNNTGCCA], 2339 (TGCCAA), and 2340 (GCCAA). Asterisks signify Benjamini-Hochberg corrected *P*-values from a Fisher's exact test that are ≤ 0.01. **c** Transcription factor footprinting analysis of differentially accessible peaks between BM- and CB-derived populations 3–5 measuring the difference in footprint depth or flanking accessibility, between BM and CB cells at each TF motif, confirming enrichment of NFI motifs in BM populations. Data are representative of two biological replicates.

time-dependent increase in: (i) *HBG* transcripts, reaching ~60% of total beta-like globin mRNA versus 10% in control, (ii) HbF+ cells (F-cells), reaching ~85% versus 15% in control, and (iii) total HbF protein, reaching ~40% compared to 3% in control (Fig. 3d–f). De-repression of *HBG* observed by NFIX knockdown reached levels equivalent to BCL11A and LRF knockdown controls in this system (Fig. 3d, e). Similarly, NFIX knockdown in HUDEP-2 cells led to robust increases in *HBG* mRNA, F-cells and total HbF protein (Fig. 3g).

**NFIX overexpression in CB cells lowers *HBG* mRNA and HbF.** Finally, to test if NFIX could silence *HBG* in cells that express elevated HbF, we ectopically expressed NFIX in CB cells, leading to NFIX mRNA and protein levels of more than 10-fold over background (Fig. 4a, b). NFIX overexpression led to: (i) reduction of *HBG* transcript levels to 40% of total beta-like globin mRNA

compared to 80% in control, (ii) reduction of F-cells to 66% compared to 96% in control, and (iii) reduction of total HbF to 31% compared to 52% in control (Fig. 4c–e). Together, these results confirm that NFIX is a potent HbF repressor.

**Discussion**
A role for NFIX in stage-specific regulation of the beta-like globin genes is supported by indirect evidence from several studies. NFI-factors were implicated in alpha-globin gene expression preference in an in vitro chicken erythrocyte system[24]. DNA methylation within the *NFIX* gene body was strongly associated with gestational age at birth in nucleated red blood cells derived from human cord blood[25,26]. In a comparison of human BM- and fetal liver-derived erythroblasts, differential DNA methylation regions were associated with NFI motifs globally and at specific CpG sites within the *NFIX* gene[18]. Most notably, a GWAS study

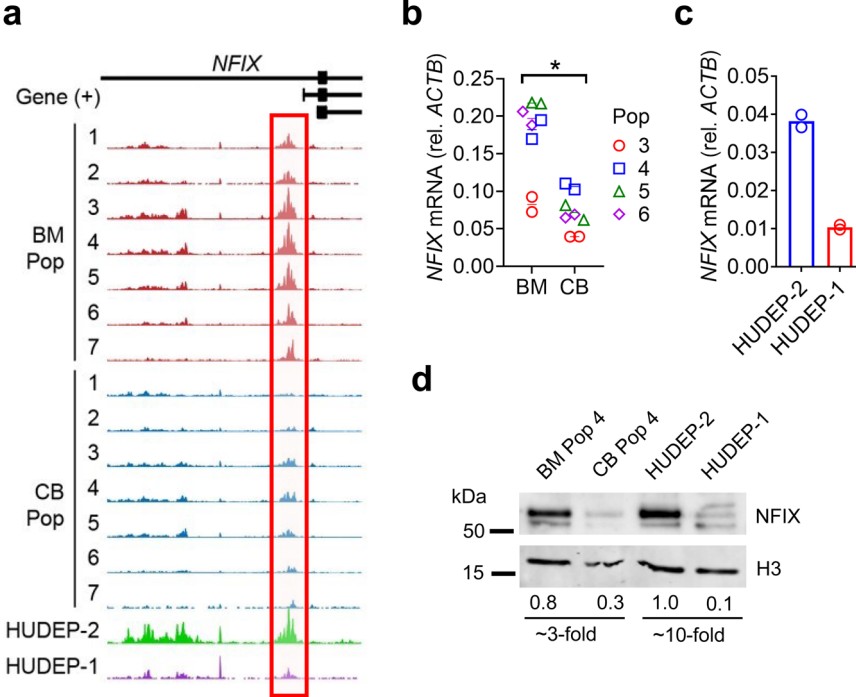

**Fig. 2 *NFIX* chromatin accessibility and expression level are reduced in cells with elevated HbF. a** ATAC-seq profiles spanning Chr19:13,122,000–13,138,000 (hg19) of sorted BM (HbF low) and CB (HbF high) cell populations showing increased chromatin accessibility at the *NFIX* promoter in BM cells (red boxes). *NFIX* splice variants NM_002501 (line 1), NM_001271044 (line 2), and NM_001365985 (line 3) are shown at the top. **b**, **c** *NFIX* mRNA quantification in BM versus CB sorted cell populations and in HUDEP-2 (HbF low) versus HUDEP-1 (HbF high). Bars represent mean of two biological replicates. Asterisk denotes statistically significant data using Student's *t*-test [*P*-value < 0.05 for BM versus CB cells (paired *t*-test, **b**)]. **d** Western blot for NFIX protein. Histone H3 served as a normalizing control. Representative immunoblots from two biological replicates are shown.

showed that SNP rs183437571, located within an intron of *NFIX* was associated with elevated HbF, albeit just below the empirical significance threshold of the study[21]. The SNP resides within a 300 kb region that includes several genes involved in erythropoiesis and erythrocyte traits, most notably KLF1, which is known to repress HbF[27,28]. KLF1 haploinsufficiency has also been suggested to contribute to hereditary persistence of fetal hemoglobin (HPFH) in two patients with microdeletion of chromosome 19p13.2–p13.12/13, a region that also includes *NFIX*[29]. Despite these data suggesting that NFIX may be involved in beta-like globin gene regulation, its direct validation as an HbF repressor had not been reported prior to our study[30].

Concurrent with our work, Qin et al. validated the role of NFIA and NFIX in HbF repression[31]. Their data suggested a dual mechanism for NFI factors in HbF repression via direct binding of NFIX/A at the *HBG* promoter and by NFIX/A-mediated modulation of BCL11A expression. Consistent with their findings for NFIX/A, we observed reduced chromatin accessibility at the *BCL11A* erythroid-specific enhancer in CB versus BM populations 3–5, with a corresponding 1.8-fold average decrease in *BCL11A* mRNA in populations 3–7 (Supplementary Fig. 6a, b). We also observed decreased chromatin accessibility at the *BCL11A* erythroid-specific enhancer upon NFIX KD in BM cells, with a corresponding 2.0-fold average reduction in *BCL11A* mRNA (Supplementary Fig. 6c, d). We did not observe remarkable changes in *ZBTB7A* chromatin accessibility or mRNA levels in the CB versus BM cell populations or following NFIX knockdown (Supplementary Fig. 6e–h). While these findings suggest NFIX may affect BCL11A levels, the delay in erythroid differentiation that we observed upon NFIX knockdown (Supplementary Fig. 5) could also contribute to the apparent reduction in *BCL11A* mRNA, since BCL11A rises during erythroid maturation[4] (Supplementary Fig. 6b).

While the HbF induction reported by Qin et al. was statistically significant when targeting NFIX, the magnitude of HbF induction was not nearly as robust as we observed using an shRNA knockdown approach. In primary cells, Qin et al. reported that CRISPR/Cas9 RNP targeting of NFIX led to approximately 10% *HBG* mRNA, 30% F-cell and 10% HbF protein levels compared to values of 60%, 85%, and 40%, respectively, in our study. An even greater discrepancy was observed in HUDEP-2 cells, where Qin et al. reported CRISPR/Cas9 knockout yielded F-cell and *HBG* mRNA values of ~5% and ~10%, respectively, compared to the ~60% and ~80% observed in our study. Additionally, Qin et al. noted that their NFIX knockout xenotransplantation experiments, which showed a phenotype identical to controls, were inconsistent with two published studies that demonstrated *Nfix* is required for hematopoietic stem and progenitor cell (HSPC) homing in mice[32,33].

Functional or hypomorphic splice variants of target genes can be created by CRISPR/Cas9-mediated genetic manipulations and stable expression of these splice variants can go undetected. For example, Poh et al. have shown that a widely used "Mettl3 knockout" cell line undergoes alternative splicing to bypass CRISPR/Cas9-induced mutations, creating a smaller but catalytically active METTL3 protein[34]. Similarly, in the case of Qin et al., the guide RNAs and validation reagents used in their NFIX knockout, which all reside within the C-terminal activation domain[35], leaves open the possibility that a functional or hypomorphic NFIX splice variant, containing a functional N-terminal DNA binding and dimerization domain, is still expressed. We recognize that the shRNA knockdown approach taken in our work has its own caveats, including potential off-target action towards related NFI factors, which could contribute to the apparent differences in our work. However, the strength of the repressive phenotype we observed when NFIX was singly overexpressed in cord blood erythroblasts, supports the notion that NFIX can act

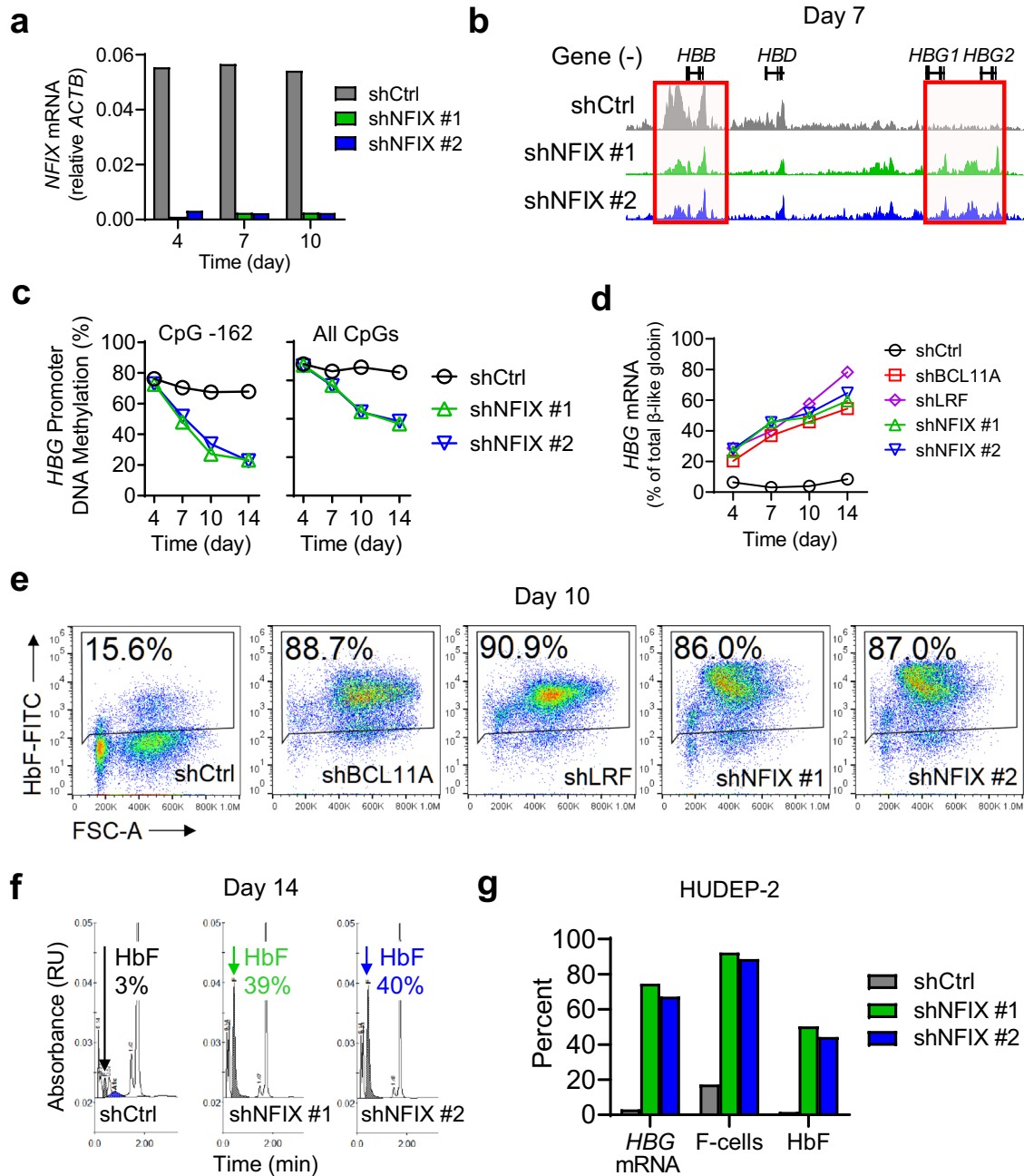

**Fig. 3 NFIX knockdown in BM and HUDEP-2 cells leads to functional changes at the *HBG* promoter and increases in *HBG* mRNA, F-cells, and HbF. a** RT-qPCR validation of NFIX knockdown in BM cells shows a 90% reduction in *NFIX* transcripts. **b** ATAC-seq profiles spanning Chr11:5,245,000–5,277,000 (hg19) of BM control and NFIX knockdown cells showing reduced chromatin accessibility at the *HBB* promoter and increased chromatin accessibility at the *HBG* promoter relative to the control cells (BM day 7 of differentiation, red boxes). **c** Percent *HBG* promoter DNA methylation at CpG -162 and an average of all 6 CpGs tested shows decreased DNA methylation in NFIX knockdown BM cells relative to the control. **d** Knockdown of NFIX in BM cells results in induction of *HBG* mRNA. **e** NFIX knockdown increases the number of F-cells (BM day 10 of differentiation). **f** HPLC chromatograms show increased absolute HbF levels in NFIX knockdown cells (BM day 14 of differentiation). **g** Knockdown of NFIX in HUDEP-2 cells increases *HBG* mRNA, F-cells, and HbF relative to the control. Data shown are representative of $N = 3$ independent experiments (**a**, **e**, **f**) and $N = 2$ independent experiments (**b**, **c**, **d**, **g**) using distinct BM and CB donors and independent HUDEP-2 transductions.

alone as a potent HbF repressor (Fig. 4). Additionally, while a residually expressed NFIX hypomorph or splice variant in Qin et al. may explain the attenuated HbF response in cells and the lack of an HSPC repopulation phenotype in mice, their electrophoretic mobility shift assay clearly shows that ectopically expressed NFIX can bind to oligonucleotides containing the NFI factor motifs present at the *HBG* promoter, implicating a direct role for NFIX in HbF repression.

NFIX is a site-specific DNA-binding transcription factor that can activate or repress genes depending on the cellular context[19]. In the mouse embryo, *Nfix* is required for proper development of the central nervous system[36], bone[37], and muscle[38], where it drives a shift from embryonic to fetal myogenesis. In humans, point mutations or deletions in the *NFIX* gene lead to Sotos syndrome, Malan syndrome, or Marshall-Smith syndrome, diseases with neurological and skeletal abnormalities[39,40]. In adult

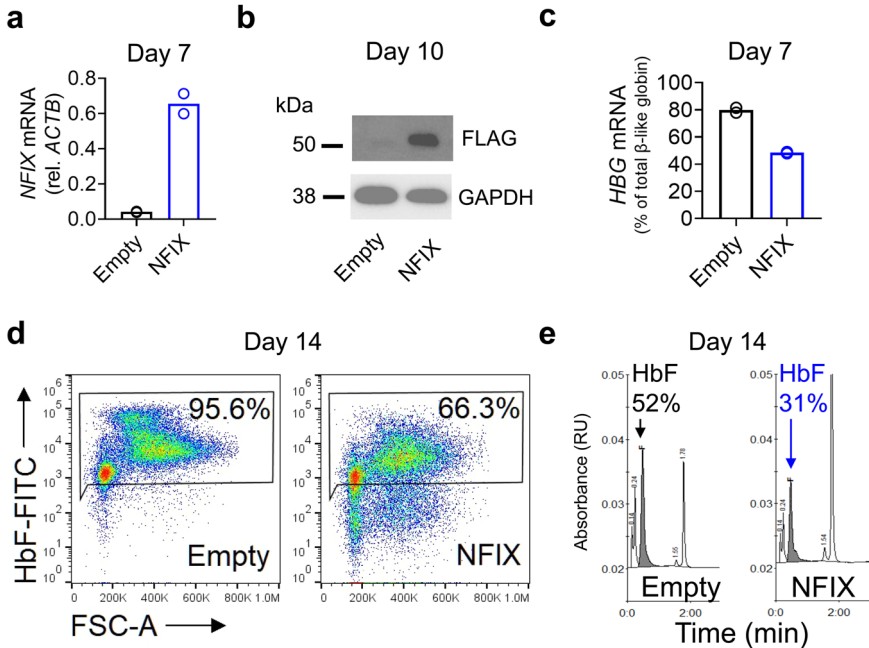

**Fig. 4 NFIX overexpression in CB cells represses HbF. a** RT-qPCR validation of *NFIX* overexpression in CB cells shows approximately 10-fold increase in *NFIX* transcripts relative to empty vector control cells. Bars represent mean of two biological replicates. **b** Western blot validation of NFIX overexpression in CB cells as determined by detection of the FLAG tag on day 10 of erythroid differentiation. **c** CB cells overexpressing NFIX have a reduction in *HBG* mRNA levels relative to other beta-like globins on day 7 of erythroid differentiation. Bars represent mean of two biological replicates. **d** CB cells overexpressing NFIX show reduced percentage of F-cells relative to empty vector control on day 14 of erythroid differentiation. **e** HPLC profiles demonstrate reduction in HbF protein in NFIX overexpressing CB cells relative to the empty vector control on day 14 of erythroid differentiation. Data are representative of $N = 2$ biological replicates using distinct CB donors and independent NFIX transductions.

mice, *Nfix* functions in the hematopoietic compartment, where it is required for stem and progenitor cell survival and hematopoietic repopulation[32]. Loss of *Nfix* also impairs myelopoiesis and enhances B-cell development[41]. In this way, NFIX is not unlike the potent HbF repressor BCL11A, which plays essential roles in hematopoiesis and neurodevelopment. In adult mice, *Bcl11a* is essential for B-cell lymphopoiesis, dendritic cell development, and maintaining the lymphoid developmental potential of early hematopoietic progenitors[42,43]. Furthermore, human genetic data has implicated a role for BCL11A in neurodevelopment[44]. Yet, despite these potential liabilities, BCL11A has become the leading therapeutic target for sickle cell disease due to the magnitude of HbF induction upon BCL11A perturbation, genetic engineering advances[45–47], and discovery of BCL11A's erythroid-specific enhancer[48].

Our findings indicate that NFIX, like BCL11A, may have a large therapeutic window, whereby partial inhibition may lead to clinically impactful HbF without affecting erythropoiesis. NFIX knockdown led to a robust HbF phenotype, with levels comparable to that of BCL11A and LRF knockdown (Fig. 3d, e), without overtly affecting red cell maturation (Supplementary Fig. 5). Similarly, maturation profiles of CB- and BM-derived erythroblasts are comparable (Fig. 1a), despite the elevated levels of HbF and lower levels of NFIX found in CB relative to BM (Fig. 2b, d). Finally, GWAS data suggests that loss of NFIX may elevate HbF without adverse effects on human health[21]. Therefore, it is likely that targeting NFIX activity in the erythroid lineage could yield therapeutic benefit without causing adverse effects, analogous to the clinically successful BCL11A shRNA^miR or *HBG* promoter binding site ablation approaches[46,47]. The most advanced clinical approach inactivates BCL11A via disruption of its erythroid-specific enhancer[45]. By analogy, NFIX may possess a yet-to-be discovered cell lineage-specific enhancer or epigenetic modification, which would allow for an additional targeting approach. The

robustness of the HbF phenotype upon NFIX knockdown and the potential tractability of targeting NFIX by genetic engineering approaches make it an important new target that could yield a therapeutic benefit to hemoglobinopathy patients. Future studies will address the mechanism by which NFIX exerts its HbF repressive function and define the regulatory elements that drive NFIX expression in adult erythroid cells.

## Methods

**Primary cell culture.** BM- or CB-derived CD34+ hematopoietic stem cells (HSCs) were purchased from AllCells, LLC, thawed according to manufacturer's instructions, and differentiated using a three-phase erythroid differentiation medium (EDM) protocol at recommended cell densities[15]. To allow for recovery and expansion of HSCs, CD34+ cells were first cultured in EDM supplemented with StemSpan™ CC100 (StemCell Technologies), but lacking hydrocortisone, EPO, SCF, and IL-3, for 3 days prior to shifting to the EDM differentiation protocol at day 0.

**HUDEP cell culture.** HUDEP-1 and HUDEP-2[16] cells were purchased under a licensing agreement from RIKEN BioResource Research Center, Japan. Cells were maintained in StemSpan™ SFEM (Stemcell Technologies) supplemented with dexamethasone (Sigma), doxycycline (Sigma), SCF (R&D Systems), and erythropoietin (R&D Systems).

**Sorting of BM- and CB-derived erythroid cell subpopulations.** BM- and CB-derived erythroid progenitor cells were differentiated using the three phase EDM. The differentiation state of the cells was determined by staining the cells with fluorescently conjugated antibodies specific to erythroid surface markers: CD36-FITC (BD Biosciences, 20 μL/test), CD71-APC (BD Biosciences, 20 μL/test) and CD235a-BV421 (BD Biosciences, 5 μL/test). The antibody co-stained cells were subjected to FACS (Sony SH800, Sony Biotechnology) and were gated based on differential expression of CD36, CD71, and CD235a on different days of erythroid differentiation: populations 1 and 2 (day 0), populations 3 and 4 (day 4), populations 5, 6, and 7 (day 10 for BM and day 11 for CB). Populations 6 and 7 for BM and CB cells were collected on different days of differentiation to account for slight differences in speed of maturation of BM and CB cells. Purity of the sorted cell populations was assessed by flow cytometry, centrifugation, and cytospin analyses.

**Cytospin analyses**. A total of 25,000–100,000 cells were used to generate cytospin slides (Shandon) and were fixed using methanol and stained with 0.08% 3′,3′-diaminobenzidine tetrahydrochloride (Sigma) solution in PBS reacted with 0.03% hydrogen peroxide (Sigma). Slides were then stained with 5% Giemsa solution (Sigma) and imaged using Axio Lab.A1 light microscope (Zeiss) at 40× magnification.

**Plasmids and lentiviral production**. MISSION® pLKO.1-puro non-mammalian shRNA plasmid (SHC002, negative control) and plasmids harboring shRNAs targeting human NFIX (TRCN0000234765, TRCN0000234767, TRCN0000234768, TRCN0000014774, and TRCN0000014776, sh#1-5 respectively) were purchased from Sigma. 293 T cells (Thermo Fisher HCL4517) were co-transfected with shRNA plasmids and pC-Pack2 lentiviral packaging plasmid mix (Cellecta) using the X-tremeGENE™ 9 DNA transfection reagent (Millipore Sigma) mixed in Opti-MEM™ (Thermo Fisher Scientific) and lentiviral particles were collected after 48 h.

**shRNA knockdown and selection**. Primary BM CD34+ cells were transduced with lentivirus harboring non-mammalian shRNA control (SHC002) or shRNAs targeting NFIX. Briefly, cells were spinoculated at $1000 \times g$ for 2 h with lentiviral particles in the presence of polybrene (Sigma). Hairpins that led to maximal knockdown of NFIX in BM cells (sh#1–2) were used to transduce HUDEP-2 cells as described above. Transduced BM and HUDEP-2 cells were plated in appropriate media and selected by addition of 2 μg/mL puromycin for 3–4 days.

**RNA isolation, cDNA preparation, and RT-qPCR**. RNA was isolated using RNeasy Mini Kit (QIAGEN) and cDNA was synthesized using SuperScript™ IV VILO™ Master Mix (Thermo Fisher Scientific). Knockdown of *NFIX* mRNA was confirmed by RT-qPCR using multiple TaqMan® Gene Expression Assays (Thermo Fisher Scientific) by normalizing to actin. Percent *HBG* mRNA was determined using a standard curve derived from plasmid dilutions harboring the *HBE, HBG, HBD,* and *HBB* coding regions. TaqMan® Probe IDs used in this study: Hs_00362215_g1 (*HBE*), Hs00361131_g1 (*HBG*), Hs_00426283_m1 (*HBD*), Hs_00747223_g1 (*HBB*), Hs_00958843_m1 (*NFIX*), Hs01093198_m1 (*BCL11A*), Hs_00252415_s1 (*ZBTB7A*), and Hs99999903_m1 (*ACTB*).

**Quantification of relative and absolute levels of HbF in BM and HUDEP-2 cells**. Control and knockdown cells were fixed with 0.05% glutaraldehyde (Sigma), permeabilized with Triton X-100 (Life Technologies), and were either stained with FITC mouse IgG₁, κ isotype control antibody (BD Biosciences, 40 μL/test) or FITC-conjugated HbF monoclonal antibody (Thermo Fisher Scientific, 10 μL/test). Stained cells were run on a Sony SH800 Cell Sorter (Sony Biotechnology) and analyzed using FlowJo™ (Becton, Dickinson and Company; 2019) to determine the percentage of F-cells. Absolute levels of HbF were determined by running lysates of $1 \times 10^6$–$3 \times 10^6$ cells on D-10 Hemoglobin Testing System (Bio-Rad) according to manufacturer's instructions.

**DNA methylation studies**. Genomic DNA from bulk populations of control and knockdown BM cells was harvested using AllPrep DNA/RNA Micro Kit (QIA-GEN). CpG methylation at positions −162, −53, −50, +6, +17, and +50 relative to the *HBG2* transcription start site[22] was measured by direct pyrosequencing after bisulfite modification and PCR amplification of genomic DNA at EpigenDx (Hopkinton, MA).

**NFIX overexpression in cord blood cells**. The CMV promoter of pLVX-CMV-IRES-Puro (Clontech) backbone vector was replaced by the *SPTA1* promoter sequence[49]. The coding sequence from NFIX transcript variant 3 (RefSeq NM_001271044.3) with a C-terminal FLAG tag was cloned into the multiple cloning site of the pLVX-SPTA1-IRES-Puro vector using SpeI/NotI (Genscript). Lentiviral particles were produced by transfecting 293 T cells as described earlier, concentrated using Lenti-X concentrator (Clontech), and viral titers were estimated using Lenti-X GoStix (Clontech) according to manufacturer's instructions. Fetal CB-derived CD34+ cells (AllCells, LLC) were transduced with concentrated lentivirus at an MOI = 1. Transduced cells were selected in the presence of 2 μg/mL puromycin for 4 days. *HBG* mRNA, F-cells, and total HbF protein in NFIX-overexpressing cells were measured as described above.

**Western blots**. For quantification of NFIX in BM, CB, HUDEP-1, and HUDPE-2 cells, nuclear and cytoplasmic fractions were separated from whole cell pellets using Nuclear Extract Kit (Active Motif®) according to manufacturer's instructions. Nuclear fractions were run on NuPAGE™ 4 to 12%, Bis-Tris gels and immuno-blotted using standard procedures and following antibodies: NFIX (Abcam Ab101341, 1:1000), H3 (Abcam Ab24834, 1:1000), GAPDH-HRP (Abcam Ab97051, 1:2000 dilution), BCL11A (Abcam Ab 191401, 1:500) and a custom LRF monoclonal antibody (Genscript, 1:3000). Western blots for NFIX overexpression studies were performed using whole cell lysates.

**ATAC-seq library construction**. ATAC-seq libraries were prepared used the Nextera DNA Library Prep Kit. A total of 50,000–100,000 cells were treated with 2.5 μL transposase for 30 min, then amplified for an initial 5 cycles using custom Illumina adapters. To determine the number of additional PCR cycles needed to amplify the libraries to 25% of the maximum qPCR fluorescence intensity, the library was quantified using SYBR Gold in a qPCR assay. The libraries were then amplified using the above-mentioned cycle numbers and sequenced on an Illumina HiSeq2000 using paired-end 40 bp reads at the Genome Technology Core at the Whitehead Institute (Cambridge, MA).

**ATAC-seq data processing and normalization**. ATAC-seq was performed on each sorted cell population from BM and CB. Adapter sequences were trimmed off the ATAC-seq reads and then the reads were aligned to human reference sequence hg19 using Bowtie2[50]. Duplicate reads were then removed before peak calling. To create a universal peak map, the 500 bp around the summit of the top 50k ATAC-seq peaks in each sample were merged together. The number of reads overlapping each peak in each sample was calculated to represent the peak score. The ATAC-seq peak scores were normalized using the varianceStabilizingTransformation (VST) function from the DESeq2 package[51]. Prior to clustering and principal component analysis (PCA), peaks were filtered using a threshold equal to the 80th percentile of the maximum peak score for each peak across all the samples.

**Defining differential ATAC-seq peaks for motif enrichment**. Peaks were grouped into three population windows: early (populations 1–2), mid (populations 3–5), and late (populations 6–7). Within each population window, peaks were filtered such that only peaks with a maximum peak score greater than the 80th percentile threshold were included for downstream analysis. For each population window differential ATAC-seq peaks were called via DESeq2, using sample source (CB or BM) and population as covariates. For motif enrichment, differential ATAC-seq peaks were defined for each direction (higher in BM and higher in CB) by FDR < 0.1 and $\log_2$(fold-change) > $\log_2$(1.25). A background set of ATAC-seq peaks were defined for each direction by $p$-value > 0.1 and $\log_2$(fold-change) < $\log_2$(1.25). The GC-content of all differential and background peaks was defined and then binned into 20 equally sized groups. 2500 background peaks per direction were sampled from the full list of background peaks using a weighted probability to match the proportion of GC-content bins in the differential peaks.

**Performing motif enrichment in differential ATAC-seq peaks**. Transcription factor (TF) binding sites were identified based on known binding motifs from various public datasets and assigned a unique Syros identifier. Motifs were drawn from the following databases: motifs based on ENCODE data[52] (http://compbio.mit.edu/encode-motifs), JASPAR[53], uniPROBE[54], and high-throughput SELEX[55]. Motifs mapping to the same TF were ranked according to the database that they originally came from: (1) ENCODE, (2) JASPAR, (3) SELEX, (4) TRANSFAC[56]. Syros motifs 2339, 2340, and 1438 reported in this work correspond to JASPAR matrix ID MA0670.1, JASPAR matrix ID MA0671.1, and the NFI motif from Jolma et al.[55], respectively. Next, FIMO[57] was used to find instances of the motifs in the genome, with a $p$-value cutoff of $10^{-4}$ and only the top 500 k instances per motif. All motifs from the curated motif database that corresponded to TFs that were expressed in any of the samples with RNA-seq were evaluated. For each motif and each direction (higher in CB or higher in BM), a Fisher's exact test was performed for association of the presence of the motif in a peak and whether the peak was differential for a GC-matched background. The estimate (odds-ratio) and $p$-value from the Fisher's exact test were evaluated to determine enrichment. ATAC-seq BM versus CB motif enrichment data are provided as Supplementary Data 1.

**Hexamer frequency to normalize for transposase sequence bias**. To account for the sequence bias of the transposase used in ATAC-seq, the frequency of all hexamer sequences in the human genome and around each motif instance was calculated according to previously described methods[17,58]. First, the frequency of each possible hexamer in the genome (hg19) excluding blacklist regions[59] was calculated. Next, for each motif in the motif database, the frequency of each hexamer at each position around the motif center (−250 bp to +250 bp) for all instances of the motif in the genome was determined. For each motif, these values were then normalized at each position such that the mean frequency of all hexamers at each position is one. The frequency of each hexamer at the ATAC-seq cut sites was then determined. The frequency of each hexamer at the ATAC-seq cut sites for each ATAC-seq sample was calculated and converted to a percent of cuts for each sample (such that the sum of all hexamers is one). This percent for each hexamer was then normalized by the genomic background frequency of that hexamer (as a percent of all hexamers such that the sum is one) to result in the normalized cut frequency of each hexamer. This was finally converted to a proportion based on the values of the other hexamers, such that the sum of the final hexamer frequency factors adds to one for each ATAC-seq sample.

**Footprint calculations**. Footprint scores were then calculated for each ATAC-seq sample and motif. For each ATAC-seq sample and motif, the frequency of cuts at each position around the motif centers (−250 bp to +250 bp) was calculated. A

background expected frequency was then calculated for each motif in each ATAC-seq sample by multiplying the motif hexamer frequency at each position around the motif (based on genomic frequencies) by the ATAC-seq cut hexamer frequency factor for that sample and normalizing such that the sum of each position around the motif is 1. Finally, the true ATAC-seq cuts at each position around each motif was normalized such that the mean is one and divided by the background expected cuts at each position in each motif (also normalized such that the mean is one). The log2 of this ratio was then used to determine the footprint scores for each motif in each sample. Lastly, a flanking accessibility (FA) and footprint depth (FPD) score were calculated for each motif in each ATAC-seq sample. The width of each motif was first determined by the width of the position weight matrix (PWM) to use in determining the regions around the motif center to use in calculating the FA and FPD scores. These regions were as follows: base region of the motif was defined the area within $(width/2 + 5)$ bp from the motif center; flank region around each motif was defined as the area from 50 bp from the motif center to the minimum of 18 bp and $(width/2 + 10)$ bp away from the motif center; background region around each motif was defined as the area 200 bp to 250 bp away from the motif center. The scores were defined as follows based on the log2 normalized ratios of cut frequencies at each position around the motif center: FPD = trimmed_mean(base region) – mean(flank region), where the trimmed_mean is the mean with the top and bottom 10% of values trimmed off; FA = mean(flank region) – mean(background region).

**Defining differential ATAC-seq peaks for cell lines and shRNA knockdowns.** Construction and data preprocessing of ATAC-seq libraries for HUDEP-1, HUDEP-2, and transduced primary CD34$^+$ BM cells were performed as described above. To increase statistical power of HUDEP-1 vs. HUDEP-2 comparisons, data from Cheng et al. (GSE157310)[20] were included in these analyses and subject to the same data preprocessing procedures. Union ATAC-seq peak sets were constructed for both the cell line samples and the shRNA knockdown samples. ATAC-seq reads were quantified within those union peak sets, and differential peaks (HUDEP-1 vs. HUDEP-2 or shRNA knockdown vs. control) were calculated via DESeq2. Dataset source was included as a covariate in the HUDEP-1 vs HUDEP-2 comparison to control for batch effects.

**Statistics and reproducibility.** The type of statistical test and *P*-values are mentioned in each figure legend along with the number of biological replicates (2-3) for each panel. Most statistical analyses were performed using a two-tailed Student's *t*-test and $P < 0.05$ was considered statistically significant. Statistical analyses performed on the ATAC-seq data are described in the figure legends and methods. Additional details can be found in the Reporting Summary linked to this paper.

**Reporting summary.** Further information on research design is available in the Nature Portfolio Reporting Summary linked to this article.

## Data availability

RNA-seq and ATAC-seq data presented in this publication have been deposited in NCBI's Gene Expression Omnibus[60] and are accessible through GEO Series accession number GSE196731 (ATAC-seq) and GSE234155 (RNA-seq). All unprocessed Western blot images are included in the Supplementary Information as Supplementary Fig. 7. Numerical data for graphs and plots are provided as Supplementary Data 2.

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

## Acknowledgements

We thank Matthew L. Eaton, Matthew G. Guenther, David A. Roth, and Eric R. Olson for helpful discussions and critical reading of the manuscript. We thank Nisha Rajagopal for data submission to NCBI GEO repository. HUDEP-1 and HUDEP-2 cells were kindly provided by Yukio Nakamura through a licensing agreement with the RIKEN Institute and the Medical and Biological Laboratories Co, Ltd.

## Author contributions

J.R.S., J.P.C., D.H.M., M.C., C.F., and B.J. designed the study, B.J., J.R.S., and M.C. performed the BM and CB sorting experiments and prepared samples for ATAC-seq experiments, C.F. and A.D. developed computational tools and performed ATAC-seq data analyses, M.C., B.J., D.H.M. performed the NFIX knockdown and overexpression experiments and associated phenotypic assays. J.R.S. and M.C. wrote the manuscript with input from all authors.

## Competing interests

M.C., C.F., B.J., A.D., and J.P.C. have an equity position in Syros Pharmaceuticals, Inc. D.H.M. and J.R.S. declare no competing interests.
