## [Peer Review File · Communications Biology]

Reviewers' comments:

Reviewer #1 (Remarks to the Author):

In this interesting manuscript, Mudit Chaand et al. provide the evidence that transcription factor NFIX is involved in fetal γ -globin repression in adult human erythroid cells.

The author compared the chromatin accessibility (by ATAC-seq) between BM and CB derived human CD34+ cells along erythroid differentiation. They found the NFI motifs were enriched in BM derived erythroid cells, suggesting the transcription factors that binding to these motifs may play a role in distinguishing the transcriptional profile between BM and CB erythroid cells. The authors further found that the promoter of NFIX, one of the transcription factors that bind to NFI motif, has increased chromatin accessibility in BM (versus CB) derived erythroid cells. This is consistent with the higher expression level of NFIX in BM erythroid cells. Interestingly, a SNP associated with high HbF that located in the NFIX gene locus has been identified in previous GWAS study.

Next, the authors test the hypothesis that NFIX represses fetal globin expression in human BM CD34+ erythroid culture by shRNA-mediated knockdown. The HbF induction after NFIX knockdown was confirmed by HPLC, F-cell staining and mRNA analysis. The authors further overexpressed NFIX in CB derived CD34+ erythroid culture and shown that NFIX overexpression reduced the level of HbF by HPLC in CB derived CD34+ erythroid culture.

Overall, the manuscript provides an interesting new finding. A few points however remain to be addressed in my opinion as detailed below.

Point-1: All the data in figure2C-G need to be repeated for three times, in different donors if the experiments performed with CD34+ cells.

Point-2: It will be very interesting to investigate the mechanism of NFIX mediated HbF repression. If there is, incorporate some analysis of NFIX ChIP-seq from the existing database.

Point-3: The gating of pop5 in figure1A looks different between BM and CB cells, please explain.

Point-4: For several of HbF staining and flow data, for example Figure2D left panel, there are two population of cells with different FSC, are the dot with lower FSC cell debris?

Reviewer #2 (Remarks to the Author):

Chaand et al. use chromatin accessibility, assessed by ATAC-seq, to identify the NFIX transcription factor as a potential repressor of the fetal beta-like globin genes HBG1/2 in adult human erythroid cells. Reactivation of HBG expression is clinically relevant in the context of beta-hemoglobinopathies. This is an interesting study; I have several comments and questions.

- 1) The choice to use cord blood as source of fetal CD34+ cells was probably based on practical considerations. A comparison with NFIX expression data obtained from human fetal liver cells would be useful (see ref 10).
2. Fig1B: It is hard to make out that these are NFI motifs. What is the consensus?
3. Fig.1D: This is not the promoter of the NFIX gene, but an intron. Which version of the human genome was used, GRCh37? What are the scales of the tracks, also for Fig.2a?
4. NFIX shRNA-mediated knockdown studies: this approach is quick but notorious for yielding false-positive results due to stress caused by lentiviral transduction and/or interferon response triggered by some shRNAs. CRISPR-mediated knockout is the current standard in the field.
5. DNA methylation analysis: how many individual molecules were sequenced, or is this a population

average of the methylation levels? In BCL11A haploinsufficiency HBG1/2 expression is high but the promoters maintain an adult methylation pattern with the exception of the -162 CpG, which was proposed as a biomarker for activity of the HBG1/2 promoters (PMID: 33938942). This should be mentioned.

6. No attempt is made to investigate the molecular mechanism of NF1X in HBG1/2 repression. NF1X ChIP-seq data and RNA-seq data (after NF1X depletion) are lacking. Does BCL11A still bind to the HBG1/2 promoters after NF1X depletion? Potential interplay with NFYA has not been investigated; it has recently been shown that NFYA activates the HBG1/2 promoters upon BCL11A removal (PMID: 33649594 and 34341563).

7. Do patients with Sotos, Malan and/or Marshall-Smith syndrome display increased HbF levels?

8. NF1X locus: two patients with microdeletion of 19p13.2-p13.12/13 deletion have been reported. The deleted region included NF1X (see suppl. fi. 1); the authors discuss that in addition to KLF1, NF1X haploinsufficiency might contribute to the high HbF levels (PMID: 27701781). This should be discussed.

9. The references need a thorough revision, they are sometimes incomplete (e.g. 13,14) or even incorrect (e.g. 25).

Reviewer #3 (Remarks to the Author):

The manuscript by Chaand et al. utilized ATAC-seq analyses in erythroid cells to develop interest in proteins that bind NF1 motifs (bound by NF1X) in erythroid cells. Knezetic and Felsenfeld (MCB 1993) and earlier biochemical and molecular biological studies had implicated NF1 as one of the transcription factors that may contribute to developmental stage-specific globin gene expression. In addition, based on human genetics, NF1X was noted previously as a potential regulator of fetal hemoglobin expression. Liao et al. (Cell Reports 2020) used ATAC-seq and Xu et al. (Dev. Cell 2012) used epigenomic mapping to establish regulated chromatin sites during erythroid differentiation, and the J. Stamatoyannopoulos group has provided extensive DNaseI data. Overall, the beta-globin locus, and, more generally, transcriptional mechanisms in erythroid cells, have been extensively studied. The current study provides solid data that re-implicates NF1 as a fetal hemoglobin regulator. The field is focused heavily on BCL11A as a fetal globin repressor, as this has created opportunities to genetically or perhaps pharmacologically downregulate BCL11A, thus increased fetal hemoglobin expression in hemoglobinopathies to achieve clinical outcomes. The striking finding with Orkin's work on BCL11A is that its depletion from erythroblasts upregulates fetal globin with almost no other changes in gene expression. This is what is so vital vis-a-vis considering BCL11A as a target. NF1X is a common transcription factor, and it would not be surprising if NF1X alterations dysregulate dozens or hundreds of genes. Since the primary contribution of the current work is to provide evidence that NF1 should be considered a target for fetal hemoglobin regulation, it is absolutely critical to assess whether NF1X impacts the genome and transcriptome broadly or in a more circumscribed manner. NF1X is broadly expressed in hematopoietic stem and progenitor cells, as well as differentiated progeny (<https://www.haemosphere.org/expression/show?geneId=ENSMUSG00000001911>).

Additional Comments:

1) Figure 1 – While NF1 motif enrichments are depicted, it would be much more informative to present a comprehensive analysis of motif enrichments.

2) Figure 2 – 2 shRNAs yielded an impressive upregulation of HBG mRNA. Since this result was not accompanied by genetic validation, it would be particularly convincing if the shRNA effect can be

rescued by re-expression of NF1X.

3) Supplementary figures – Much of this content is essential for the study, and only presenting the two primary figures does not seem to be ideal.

4) Suppl. Fig. 2 – Showing positive and negative control loci would facilitate interpretation of the globin locus data, and conspicuously missing is the alpha globin locus.

5) Suppl. Fig. 4 – The apparent differential expression is not supported by statistics, at least not in the legend or denoted in the figure. B, The Western blot data should be quantified to provide evidence for data reproducibility.

6) Suppl. Fig. 5 – The shRNA efficacy appears to be nearly 100%. It is not evident how such a high efficiency was achieved, as it is much more common to achieve 50-70% reductions in erythroid cells with efficacious shRNAs. Also, see the note above about rescue.

7) Suppl. Fig. 6 – No quantitative analysis supported by statistics was included to address this important point.

8) Suppl. Fig. 7 - No quantitative analysis supported by statistics was included to address this important point.

Response to Reviewers: COMMSBIO-22-0794-T

Major comments across reviewers:

A publication by Qin *et al.* in Nature Genetics (PMID 35618846) in June 2022 reported the role of NFI-factors in fetal hemoglobin silencing in adult erythroid cells. In this study, the authors performed a CRISPR-based screen in HUDEP-2 cells targeting DNA-binding domains of human transcription factors to identify NFIA and NFIX as fetal hemoglobin repressors. This study also addressed some of the concerns proposed by the referees of our manuscript (see below).

1. Reviewer 1, comment 2 and Reviewer 2, comment 6: RNA-seq studies upon NFIX knockdown and NFIX mechanism of action studies.

Qin *et al.*¹ performed a comprehensive RNA-seq analysis to identify genes that are differentially expressed upon genetic perturbation of NFIA and NFIX in HUDEP-2 cells and concluded that NFIX and NFIA promote an adult type gene expression signature (Figure 3 of Qin *et al.*). To understand the mechanism of action of NFIA and NFIX, the authors performed CUT&RUN studies in HUDEP-2 and primary cells and observed NFIX and NFIA occupancy at a BCL11A intronic enhancer (Figure 4 of Qin *et al.*). In follow-up ATAC-seq studies, the authors detected decreased chromatin accessibility at +58 and +62 *BCL11A*-specific erythroid enhancers (Figure 5 of Qin *et al.*) suggesting that NFIA/NFIX support BCL11A expression in adult erythroid cells to exert their repressive function on *HBG1/2* genes. Finally, they present electrophoretic mobility shift assay data showing that NFIX and NFIA can bind to DNA sequences present at the *HBG* promoter containing NFI factor binding motifs. Together, they posit a dual mechanism for NFI factors in HbF repression, through direct binding at the *HBG* promoter and by regulation of BCL11A expression. We have included text referencing their work and the proposed mechanism in our discussion along with additional data to support the role of NFIX-mediated BCL11A regulation (supplemental Figure 7).

2. Reviewer 2, comment 4: CRISPR mediated knockout of NFIX.

We agree with the reviewer that an shRNA approach can yield false positives. To address this concern, we utilized multiple hairpins targeting different regions of the NFIX gene. Multiple unique hairpins yielding similar HbF phenotypes help to mitigate concerns of false positives. Additionally, we performed NFIX overexpression experiments in cord blood cells to show that NFIX can lower HbF levels. Used in combination, shRNA knockdown and ectopic overexpression of NFIX significantly lowers the risk of a false positive phenotype.

While CRISPR-mediated knockout is standard in the field it is technically challenging for targets with more than 2 copies, as is the case for NFIX in HUDEP-2 cells. NFIX is encoded within chromosome 19 which has 3 copies in HUDEP-2 cells. Reviewer 2's concerns have been addressed by Qin *et al.* where the authors reported a statistically significant increase in *HBG* mRNA and HbF protein in CRISPR-mediated NFIX knockout in HUDEP-2 cells and primary cells, respectively (Figures 1e and 2 b-c of Qin *et al.*).

Interestingly, while the HbF induction observed by Qin *et al.* is statistically significant upon NFIX knockout, the magnitude of HbF induction is not nearly as robust as observed by using shRNA knockdown in our study. Since NFIX has been observed in the GWAS by Danjou *et al.*² (PMID: 26366553), we would have expected the NFIX CRISPR knockout approach by Qin *et al.* to yield a much stronger HbF phenotype, similar to that observed in our study. Additionally, the authors noted that their NFIX knockout xenotransplantation experiments were inconsistent with two previous reports that Nfix is required for hematopoietic stem and progenitor cell (HSPC) homing in mice (Extended Figure 3 of Qin *et al.* and PMID: 24041575 and PMID: 29430853)^{1,3,4}. Furthermore, they report that only 7 genes were changed in RNA-seq upon NFIX knockout in HUDEP-2 cells. Together, these results suggest that the NFIX knockout in Qin *et al.* may still contain partially functional NFIX.

While CRISPR is a powerful tool, it is not without caveats. Functional or hypomorphic splice variants of target genes can be created by CRISPR-based methods and stable expression of these splice variants can go undetected. For example, Poh *et al.* have shown that a widely used “Mettl3 knockout” cell line undergoes alternative splicing to bypass CRISPR/Cas9-induced mutations, creating a smaller but catalytically active METTL3 protein (PMID: 35853000)⁵.

In the case of Qin *et al.*, the construction and validation approach used for their NFIX knockouts leaves open the possibility that they have created a hypomorphic NFIX variant containing a functional N-terminal DNA binding and dimerization domain but lacking the C-terminal transactivation domain. The sgRNAs used to generate, and the antibody used to validate, their NFIX knockout both target regions well downstream of the DNA binding and dimerization domain, thereby precluding the detection of a possible hypomorphic form of NFIX (mapping of Qin *et al.* reagents at the NFIX locus available upon request).

While we suggest the presence of hypomorphic NFIX to explain their attenuated HbF response in cells and lack of a HSPC repopulation phenotype in mice, we also recognize that the shRNA knockdown approach taken in our work has its own caveats, including potential off-target action, that could contribute to the apparent differences in our work. However, the strength of the repressive phenotype we observe when NFIX was singly overexpressed in cord blood erythroblasts, supports the notion that NFIX can act alone as a potent HbF repressor (Figure 2g, supplemental Figure 8).

We have added text to the manuscript discussing the shRNA/CRISPR discrepancy between our work and Qin *et al.*, as it will be an important point for other investigators to arbitrate in future studies and may offer insights into the NFIX’s mechanism of action in *HBG* repression.

Responses to individual reviewers' comments:

Reviewer #1 (Remarks to the Author):

In this interesting manuscript, Mudit Chaand et al. provide the evidence that transcription factor NFIX is involved in fetal γ -globin repression in adult human erythroid cells.

The author compared the chromatin accessibility (by ATAC-seq) between BM and CB derived human CD34+ cells along erythroid differentiation. They found the NFI motifs were enriched in BM derived erythroid cells, suggesting the transcription factors that binding to these motifs may play a role in distinguishing the transcriptional profile between BM and CB erythroid cells. The authors further found that the promoter of NFIX, one of the transcription factors that bind to NFI motif, has increased chromatin accessibility in BM (versus CB) derived erythroid cells. This is consistent with the higher expression level of NFIX in BM erythroid cells. Interestingly, a SNP associated with high HbF that located in the NFIX gene locus has been identified in previous GWAS study.

Next, the authors test the hypothesis that NFIX represses fetal globin expression in human BM CD34+ erythroid culture by shRNA-mediated knockdown. The HbF induction after NFIX knockdown was confirmed by HPLC, F-cell staining and mRNA analysis. The authors further overexpressed NFIX in CB derived CD34+ erythroid culture and shown that NFIX overexpression reduced the level of HbF by HPLC in CB derived CD34+ erythroid culture.

Overall, the manuscript provides an interesting new finding. A few points however remain to be addressed in my opinion as detailed below.

#	Referee Comments	Authors' Responses
1	All the data in figure2C-G need to be repeated for three times, in different donors if the experiments performed with CD34+ cells.	Experiments in panels 2C-E were repeated three times using CD34+ cells from 3 distinct donors as reflected in the figure legend. Experiments in panels 2F and G were performed in HUDEP-2 cells using the same shRNAs as CD34+ experiments. We respectfully maintain that two bioreplicates in HUDEP-2 cells are sufficient to further corroborate the results from experiments using primary CD34+ cells.
2	It will be very interesting to investigate the mechanism of NFIX mediated HbF repression. If there is, incorporate some analysis of NFIX ChIP-seq from the existing database.	Qin et al. address this in their publication (PMID 35618846). Discussed in detail above and in text added to the revised manuscript (lines 140-179 of Main Text).
3	The gating of pop5 in figure1A looks different between BM and CB cells, please explain.	We observed slight differences in speed of maturation between BM and CB cells. This is typical for primary cell cultures using CD34+ cells from different donors/lots. Populations 5-7 were collected on Day 10 of erythroid differentiation for BM cells and Day 11 for CB cells. Gates 5-7 for CB cells on Day 11 were drawn

		to best represent the distribution of BM cells on Day 10 and collect stage-matched cells for downstream ATAC-seq comparisons. These details are now included in the Methods section of the revised manuscript (lines 20-22 of Supplement).
4	For several of HbF staining and flow data, for example Figure 2D left panel, there are two population of cells with different FSC, are the dot with lower FSC cell debris?	The FSC-low population does not reflect cell debris. The debris (FSC-A low and SSC-A low) is not included in the analysis as shown in Supplemental Figure 1. As the erythroid cells differentiate, their nucleus-cytoplasm (N/C) ratio gets smaller and they eventually lose their nucleus. The FSC-low population contains the most mature erythroid cells, likely enucleated reticulocytes.

Reviewer #2 (Remarks to the Author):

Chaand et al. use chromatin accessibility, assessed by ATAC-seq, to identify the NFIX transcription factor as a potential repressor of the fetal beta-like globin genes HBG1/2 in adult human erythroid cells. Reactivation of HBG expression is clinically relevant in the context of beta-hemoglobinopathies. This is an interesting study; I have several comments and questions.

#	Referee Comments	Authors' Responses
1	The choice to use cord blood as source of fetal CD34+ cells was probably based on practical considerations. A comparison with NFIX expression data obtained from human fetal liver cells would be useful (see ref 10).	Since human fetal liver cells are not commercially available, we used CD34+ cord blood cells instead for the hypothesis generating experiment to identify novel fetal hemoglobin regulators.
2	Fig1B: It is hard to make out that these are NFI motifs. What is the consensus?	We have modified the figure to make the motif sequences more legible and have also added them to the figure legend (lines 265-266 of Main Text).
3	Fig.1D: This is not the promoter of the NFIX gene, but an intron. Which version of the human genome was used, GRCh37? What are the scales of the tracks, also for Fig.2a?	GRCh37 version of the human genome was used for alignment. The chromatin accessibility peak we reference resides at the promoter of some NFIX splice variants, including NM_001271044. We have modified the legend to denote that the figure represents RefSeq variants NM_002501 (first line), NM_001271044 (second line) and NM_001365985 (third line). We

		have also added text to the methods indicating that the coding region of NM_001271044 was used for NFIX overexpression experiments (line 63 of Supplement). A scale bar denoting base pairs has been added.
4	NFIX shRNA-mediated knockdown studies: this approach is quick but notorious for yielding false-positive results due to stress caused by lentiviral transduction and/or interferon response triggered by some shRNAs. CRISPR-mediated knockout is the current standard in the field.	Qin et al. has used CRISPR in their publication (PMID 35618846) to validate NFIX. Additionally, our NFIX overexpression expression experiments that show suppression of HbF dramatically reduce the possibility of a false positive. These points are discussed in detail above and additional discussion around this topic has been added to the revised manuscript (lines 140-179 of Main Text).
5	DNA methylation analysis: how many individual molecules were sequenced, or is this a population average of the methylation levels? In BCL11A haploinsufficiency HBG1/2 expression is high but the promoters maintain an adult methylation pattern with the exception of the -162 CpG, which was proposed as a biomarker for activity of the HBG1/2 promoters (PMID: 33938942). This should be mentioned.	Methylation analyses were performed using a bulk cell population and we have made a note of this in the Methods section (line 57 of Supplement). In the revised manuscript, we have modified the main text to highlight CpG -162 as a biomarker for activity and have added the reference as suggested by the reviewer (lines 110-111 of Main Text).
6	No attempt is made to investigate the molecular mechanism of NFIX in HBG1/2 repression. NFIX ChIP-seq data and RNA-seq data (after NFIX depletion) are lacking. Does BCL11A still bind to the HBG1/2 promoters after NFIX depletion? Potential interplay with NFYA has not been investigate; it has recently been shown that NFY activates the HBG1/2 promoters upon BCL11A removal PMID: 33649594 and 34341563).	Qin et al. address this in their publication (PMID 35618846). Discussed in detail above and an additional analysis supporting a role for NFIX in the repression of BCL11A has been presented in the revised manuscript (lines 140-179 of Main Text and Supplemental Figure S7).
7	Do patients with Sotos, Malan and/or Marshall-Smith syndrome display increased Hbf levels?	This has not been reported to the best of our knowledge.
8	NFIX locus: two patients with microdeletion of 19p13.2-p13.12/13 deletion have been reported. The deleted region	We appreciate the reviewer bringing this to our attention,

	included NF1X (see suppl. fi. 1); the authors discuss that in addition to KLF1, NF1X haploinsufficiency might contribute to the high HbF levels (PMID: 27701781). This should be discussed.	we have edited our discussion in the revised manuscript to state this finding and have added the corresponding reference (lines 136-138 of Main Text).
9	The references need a thorough revision, they are sometimes incomplete (e.g. 13,14) or even incorrect (e.g. 25).	We have reviewed all the references and have made corrections as needed.

Reviewer #3 (Remarks to the Author):

The manuscript by Chaand et al. utilized ATAC-seq analyses in erythroid cells to develop interest in proteins that bind NF1 motifs (bound by NF1X) in erythroid cells. Knezetic and Felsenfeld (MCB 1993) and earlier biochemical and molecular biological studies had implicated NF1 as one of the transcription factors that may contribute to developmental stage-specific globin gene expression. In addition, based on human genetics, NF1X was noted previously as a potential regulator of fetal hemoglobin expression. Liao et al. (Cell Reports 2020) used ATAC-seq and Xu et al. (Dev. Cell 2012) used epigenomic mapping to establish regulated chromatin sites during erythroid differentiation, and the J. Stamatoyannopoulos group has provided extensive DNaseI data. Overall, the beta-globin locus, and, more generally, transcriptional mechanisms in erythroid cells, have been extensively studied. The current study provides solid data that re-implicates NF1 as a fetal hemoglobin regulator. The field is focused heavily on BCL11A as a fetal globin repressor, as this has created opportunities to genetically or perhaps pharmacologically downregulate BCL11A, thus increased fetal hemoglobin expression in hemoglobinopathies to achieve clinical outcomes. The striking finding with Orkin's work on BCL11A is that its depletion from erythroblasts upregulates fetal globin with almost no other changes in gene expression. This is what is so vital vis-a-vis considering BCL11A as a target. NF1X is a common transcription factor, and it would not be surprising if NF1X alterations dysregulate dozens or hundreds of genes. Since the primary contribution of the current work is to provide evidence that NF1 should be considered a target for fetal hemoglobin regulation, it is absolutely critical to assess whether NF1X impacts the genome and transcriptome broadly or in a more circumscribed manner. NF1X is broadly expressed in hematopoietic stem and progenitor cells, as well as differentiated progeny (<https://www.haemosphere.org/expression/show?genelid=ENSMUSG00000001911>).

#	Referee Comments	Authors' Responses
1	Figure 1 – While NF1 motif enrichments are depicted, it would be much more informative to present a comprehensive analysis of motif enrichments.	Table S1 contains a comprehensive analysis of motif enrichments.
2	Figure 2 – 2 shRNAs yielded an impressive upregulation of HBG mRNA. Since this result was not accompanied by genetic validation, it would be particularly convincing if the shRNA effect can be rescued by re-expression of	This is a great suggestion but poses technical challenges for execution. Our NF1X shRNA knockdown lentiviral vector and our NF1X overexpression lentiviral

	NF1X.	vector have the same puromycin resistance cassette. While resistance cassettes can be modified for the two backbone vectors, serial transductions and selection can be challenging. Moreover, our NFIX overexpression experiments (Figure 2g and Supplemental Figure 8) demonstrate that NFIX overexpression in HbF-high CB cells leads to a significant reduction in HbF, corroborating the role of NFIX as a negative regulator of HbF. Please also see our discussion above, and accompanying text added to the revised manuscript, addressing the caveats of CRISPR in the context of the phenotype observed by Qin et al. when using CRISPR to knockout NFIX (lines 140-179 of Main Text).
3	Supplementary figures – Much of this content is essential for the study, and only presenting the two primary figures does not seem to be ideal.	We agree with the reviewer. Our original manuscript was formatted and submitted as a Letter to Nature Genetics, which was then automatically transferred to Communications Biology after editor review. Should our revised manuscript be accepted, and at the discretion of the editor, we propose a format with 4 primary figures and 6 supplemental figures. We propose that Supplemental Figure 4 (NFIX levels in fetal and adult state cells), Supplemental Figure 5 (NFIX KD validation), and Supplemental Figure 8A-C (NFIX overexpression in cord blood cells) are moved to the primary figures.
4	Suppl. Fig. 2 – Showing positive and negative control loci would facilitate interpretation of the globin locus data, and conspicuously missing is the alpha globin locus.	Attached to this rebuttal we have included the chromatin accessibility profiles for several control loci (Figure R1). KLF1 and GATA1 were included to control for regions that should have similar accessibility profiles in cord blood and bone marrow lineages. GAPDH was included as a control for regions that should have similar accessibility profiles across cell population and lineages. Alpha globin gene regulation is not the focus of our paper and therefore, we purposefully excluded it from this

		manuscript. However, we have included the alpha globin locus, which shows comparable accessibility within cord blood and bone marrow populations, in Figure R1 for review. With respect to ATAC-seq controls included in the manuscript, we maintain that the HUDEP-1 and HUDEP-2 tracks are sufficient as chromatin accessibility controls for CB and BM cells, since they predominantly express HBG and HBB, respectively. All ATAC-seq data will be available to the scientific community via the NCBI SRA portal to independently validate our observations and query other genetic loci of interest.
5	Suppl. Fig. 4 – The apparent differential expression is not supported by statistics, at least not in the legend or denoted in the figure. B, The Western blot data should be quantified to provide evidence for data reproducibility.	We have run a statistical test on the data in Supplemental Figure 4A to confirm that the observed differences are significant. The type of test and resulting P-value have been added to the figure and the figure legend (lines 257-262 of Supplement). We quantified the protein expression NFIX data in Figure 4B to show approximately 3-fold reduction in CB cells vs. BM cells and approximately 10-fold reduction in HUDEP-1 cells vs. HUDEP-2 cells.
6	Suppl. Fig. 5 – The shRNA efficacy appears to be nearly 100%. It is not evident how such a high efficiency was achieved, as it is much more common to achieve 50-70% reductions in erythroid cells with efficacious shRNAs. Also, see the note above about rescue.	In our first experiment to knockdown NFIX, we used 5 different shRNAs to target NFIX (sh#1-5, Methods). We observed a range of KD efficiencies (54-89% by mRNA level). We prioritized replicate experiments with shRNAs that led to the most robust NFIX knockdown as determined by RT-qPCR and Western blot while having the least effect on cell viability and erythroid differentiation as determined by erythroid surface marker staining. For supplemental figure 5, representative data from two to three biological replicates are shown. The same shRNAs in other replicates showed KD efficiency of 70-90%.
7	Suppl. Fig. 6 – No quantitative analysis supported by	We did not perform a quantitative

	statistics was included to address this important point.	analysis of enucleated cells and respectfully maintain that it is not required to highlight the observation that despite a slight delay in erythroid differentiation, the NFIX KD cells are able to enucleate. Our result has been corroborated by the CRISPR experiments of Qin et al., who observed no overt changes in cell viability or erythroid maturation upon CRISPR knockout of NFIX and/or NFIA.
8	Suppl. Fig. 7 - No quantitative analysis supported by statistics was included to address this important point.	We appreciate this comment by the reviewer and agree it is an important point, especially considering the findings of Qin et al., where they show NFI factors appear to elevate BCL11A levels by acting at the erythroid specific enhancer. Since our Western blot was from a single experiment in HUDEP-2 cells, we decided to remove this data entirely and take a new approach utilizing our existing data in primary cells. We queried the chromatin accessibility profiles at the BCL11A and ZBTB7A loci and looked at the mRNA levels of these genes in the CB versus BM and in the NFIX knockdown experimental series. Consistent with the findings of Qin et al., we observed reduced levels of chromatin accessibility at the intronic BCL11A erythroid-specific enhancer in CB versus BM populations 3-5, with a corresponding 1.8-fold decrease in BCL11A mRNA (new Supplemental Figure 7A-B, please see below). We also observed increased chromatin accessibility at the BCL11A erythroid-specific enhancer upon NFIX knockdown in BM cells, with a corresponding 2.0-fold reduction in BCL11A mRNA (new Supplemental Figure 7C-D). Also consistent with Qin et al., we did not observe remarkable changes in ZBTB7A chromatin accessibility and mRNA levels in the CB versus BM or following NFIX knockdown (Supplemental Figure 7E-H).

		We have included new text, figures, and discussion in the revised manuscript detailing these findings.
9	General comment from reviewer we would like to address: “NF1X is a common transcription factor, and it would not be surprising if NF1X alterations dysregulate dozens or hundreds of genes. Since the primary contribution of the current work is to provide evidence that NF1 should be considered a target for fetal hemoglobin regulation, it is absolutely critical to assess whether NF1X impacts the genome and transcriptome broadly or in a more circumscribed manner. NF1X is broadly expressed in hematopoietic stem and progenitor cells, as well as differentiated progeny (https://www.haemosphere.org/expression/show?qeneld=ENSMUSG00000001911)	We agree with the reviewer that NF1X is a common transcription factor that would be expected to regulate many genes beyond fetal hemoglobin. However, we respectfully disagree that defining its potential regulatory role across other lineages is within the scope of this manuscript, nor does absence of this knowledge disqualify NF1X as a potential therapeutic target. As example, like NF1X, BCL11A is expressed in a variety of hematopoietic cells at significant levels. BCL11A was discovered to be a potent fetal hemoglobin repressor in 2008, yet its erythroid-specific enhancer was not discovered until 2013. It is the discovery and refinement of the erythroid-specific enhancer that has made BCL11A a tractable target for some of the exciting and successful gene modification approaches we see today. Importantly, like BCL11A, NF1X may possess a yet-to-be discovered erythroid-specific enhancer which would make it an exciting and tractable target for CRISPR mediated approaches.

Figure R1. ATAC-seq profiles at control loci.

Chromatin accessibility at the KLF1 locus changes similarly in BM and CB cells as expected.

GATA1 accessibility changes similarly in BM and CB cells with lower accessibility in early stage populations and greater accessibility in later stage populations.

Chromatin accessibility at the GAPDH control locus is similar between BM and CB populations.

Chromatin accessibility at the alpha globin gene cluster is similar between the BM and CB cell populations.

Supplemental Figure 7

Supplemental Figure 7. Cells with higher levels of NFIX have lower levels of BCL11A mRNA. a, ATAC-seq profiles of sorted BM and CB populations at the *BCL11A* locus showing lower chromatin accessibility in populations 3-5 of CB cells (higher NFIX) relative to BM (lower NFIX). Intronic *BCL11A* erythroid-specific enhancers at +55 kb, +58 kb and +62 kb relative to the transcription start site are boxed.

b, Corresponding mRNA samples show reduced BCL11A expression levels in CB. **c**, ATAC-seq profiles of NFIX knockdown BM cells at day 4 of differentiation show reduced chromatin accessibility at the *BCL11A* enhancer (boxed) upon NFIX knockdown relative to control. **d**, Matched mRNA samples from NFIX knockdown in BM cells on days 4, 7 and 10 of differentiation show reduced *BCL11A* levels relative to control. **e-h**, *ZBTB7A* chromatin accessibility profiles and mRNA levels in the same experiments as described in **a-d**. Statistical significance of data shown from two biological replicates was determined using a paired Student's *t*-test. Asterisks denote $P < 0.05$; n.s., not significant.

References

1. Qin, K. *et al.* Dual function NFI factors control fetal hemoglobin silencing in adult erythroid cells. *Nat. Genet.* **54**, 874–884 (2022).
2. Danjou, F. *et al.* Genome-wide association analyses based on whole-genome sequencing in Sardinia provide insights into regulation of hemoglobin levels. *Nat. Genet.* **47**, 1264–1271 (2015).
3. Holmfeldt, P. *et al.* Nfix is a novel regulator of murine hematopoietic stem and progenitor cell survival. *Blood* **122**, 2987–2996 (2013).
4. Hall, T. *et al.* Nfix Promotes Survival of Immature Hematopoietic Cells via Regulation of c-Mpl. *Stem Cells* **36**, 943–950 (2018).
5. Poh, H. X., Mirza, A. H., Pickering, B. F. & Jaffrey, S. R. Alternative splicing of METTL3 explains apparently METTL3-independent m6A modifications in mRNA. *PLoS Biol.* **20**, 1–25 (2022).

Reviewers' comments:

Reviewer #2 (Remarks to the Author):

In the revised version the authors have addressed all my queries adequately.

Reviewer #3 (Remarks to the Author):

I have re-reviewed the manuscript by Chaand et al. Several revisions have appropriately addressed prior concerns. Other important recommendations were not pursued, and appropriate revisions were not implemented.

Genetic rescue with NFIX – this is important to validate the shRNA data.

Incorporating rigorous supplementary figures as primary figures – Whereas the authors indicate they will do this if the manuscript is accepted, it has not been done.

Point 7, suppl 6 – The authors still did not conduct the recommended quantitative analysis.

Point 9 – The authors still did not investigate the broader actions of NFIX, which may preclude its theoretical utility as a target for modulation of hemoglobin switching.

Key references were provided to establish an appropriate background for the NFIX work, and apparently this was ignored.

Response to Reviewers: COMMSBIO-22-0794-B

#	Referee comments	Authors' Responses
1	Genetic rescue with NFIX – this is important to validate the shRNA data.	We respectfully maintain that while this is a great suggestion, it poses technical challenges for execution, as described in detail in our previous rebuttal. While the editor agrees that a genetic rescue experiment would have strengthened the data, they have accepted the technical limitations we have described, so we have not conducted this experiment and focused on addressing the other concerns that were raised.
2	Incorporating rigorous supplementary figures as primary figures – Whereas the authors indicate they will do this if the manuscript is accepted, it has not been done.	We have reformatted the manuscript to include 4 main figures and 6 supplemental figures. A new Figure 2 has been created that contains the data from Figure 1D and the previous Supplemental Figure 4 (NFIX levels in fetal and adult state cells). A new Figure 3 has been created that includes previous Supplemental Figure 5A (NFIX KD validation in BM cells) along with previous Figure 2A-F. A new Figure 4 has been created that includes new RT-qPCR validation data of ectopic NFIX expression and all the data from previous Supplemental Figure 8A-C (NFIX overexpression in cord blood cells). The remaining data that were not moved to the main figures have been renumbered appropriately in the supplement.
3	Point 7, suppl 6 – The authors still did not conduct the recommended quantitative analysis.	We have included the flow cytometry plots to show the differentiation profile on Day 14 and have included low magnification images containing multiple enucleated cells to show that NFIX knockdown does not have a remarkable effect on terminal erythroid differentiation (Supplemental Figure 5). Flow gates C and D on Day 14 provide an estimate of reticulocytes in control versus NFIX knockdown cells.
4	Point 9 – The authors still did not investigate the broader actions of NFIX, which may preclude its theoretical utility as a target for modulation of hemoglobin switching.	We appreciate this comment and have discussed this point in greater detail in lines 185-214.
5	Key references were provided to establish an appropriate background for the NFIX work, and apparently this was ignored.	We have included additional background as recommended in lines 131-134.